# A probabilistic population code based on neural samples

**Sabyasachi Shivkumar**$^*$**, Richard D. Lange**$^*$**, Ankani Chattoraj**$^*$**, Ralf M. Haefner**
Brain and Cognitive Sciences, University of Rochester
{sshivkum, rlange, achattor, rhaefne2}@ur.rochester.edu

## Abstract

Sensory processing is often characterized as implementing probabilistic inference: networks of neurons compute posterior beliefs over unobserved causes given the sensory inputs. How these beliefs are computed and represented by neural responses is much-debated (Fiser et al. 2010, Pouget et al. 2013). A central debate concerns the question of whether neural responses represent samples of latent variables (Hoyer & Hyvarinnen 2003) or parameters of their distributions (Ma et al. 2006) with efforts being made to distinguish between them (Grabska-Barwinska et al. 2013). A separate debate addresses the question of whether neural responses are proportionally related to the encoded probabilities (Barlow 1969), or proportional to the logarithm of those probabilities (Jazayeri & Movshon 2006, Ma et al. 2006, Beck et al. 2012). Here, we show that these alternatives – contrary to common assumptions – are not mutually exclusive and that the very same system can be compatible with all of them. As a central analytical result, we show that modeling neural responses in area V1 as samples from a posterior distribution over latents in a linear Gaussian model of the image implies that those neural responses form a linear Probabilistic Population Code (PPC, Ma et al. 2006). In particular, the posterior distribution over some experimenter-defined variable like "orientation" is part of the exponential family with sufficient statistics that are linear in the neural sampling-based firing rates.

## 1 Introduction

In order to guide behavior, the brain has to infer behaviorally relevant but unobserved quantities from observed inputs in the senses. Bayesian inference provides a normative framework to do so; however, the computations required to compute posterior beliefs about those variables exactly are typically intractable. As a result, the brain needs to perform these computations in an approximate manner. The nature of this approximation is unclear with two principal classes having emerged as candidate hypotheses: parametric (variational) and sampling-based [8, 20].

In the first class, neural responses are interpreted as the parameters of the probability distributions that the brain computes and represents. The most popular members of this class are Probabilistic Population Codes (PPCs, [13, 4, 3, 2, 21, 19]). Common PPCs are based on the empirical observation that neural variability is well-described by an exponential family with linear sufficient statistics. Applying Bayes' rule to compute the posterior probability, $p(s|\mathbf{r})$, over some task-relevant scalar quantity, $s$, from the neural population response, $\mathbf{r}$, one can write [2]:

$$p(s|\mathbf{r}) \; \propto \; g(s) \exp\left[\mathbf{h}(s)^\top \mathbf{r}\right] \tag{1}$$

where each entry of $\mathbf{h}(s)$ represents a stimulus-dependent kernel characterizing the contribution of each neuron's response to the distribution, and $g(s)$ is some stimulus-dependent function that

---

$^*$Equal contribution

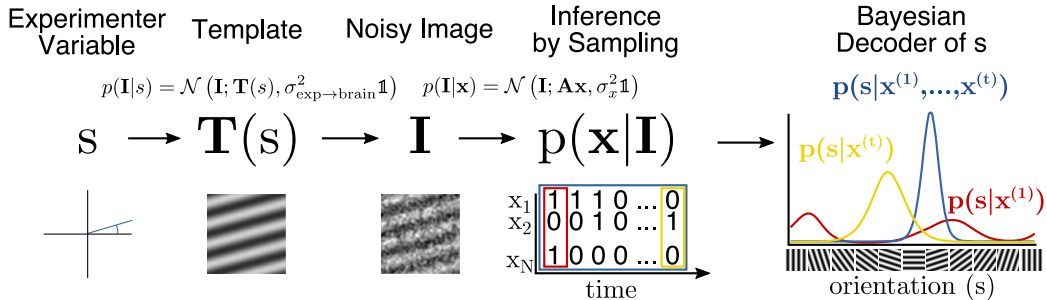

Figure 1: General setup: Our model performs sampling-based inference over $\mathbf{x}$ in a probabilistic model of the image, $\mathbf{I}$. In a given experiment, the image is generated according to the experimenter's model that turns a scalar stimulus $s$, e.g. orientation, into an image observed by the brain. The samples drawn from the model are then probabilistically "decoded" in order to infer the implied probability distribution over $s$ from the brain's perspective. While the samples shown here are binary, our derivation of the PPC is agnostic to whether they are binary or continuous, or to the nature of the brain's prior over $\mathbf{x}$.

is independent of $\mathbf{r}$. Importantly, the neural responses, $\mathbf{r}$, are linearly related to the logarithm of the probability rather than the probability itself. This has been argued to be a convenient choice for the brain to implement important probabilistic operations like evidence integration over time and cues using linear operations on firing rates [2]. In addition, PPC-like codes are typically "distributed" since the belief over a single variable is distributed over the activity of many neurons, and different low-dimensional projections of those activities may represent beliefs over multiple variables simultaneously [19]. Furthermore, because $s$ is defined by the experimenter and not *explicitly* inferred by the brain in our model we call it "implicit."

In the second class of models, instead of representing parameters, neural responses are interpreted as samples from the represented distribution. First proposed by Hoyer & Hyvarinnen (2003), this line of research has been elaborated in the abstract showing how it might be implemented in neural circuits [7, 18, 5] as well as for concrete generative models designed to explain properties of neurons in early visual cortex [14, 15, 24, 12, 16, 10]. Here, each neuron (or a subset of principal neurons), represents a single latent variable in a probabilistic model of the world. The starting point for these models is typically a specific generative model of the inputs which is assumed to have been learnt by the brain from earlier sensory experience, effectively assuming a separation of time-scales for learning and inference that is empirically justified at least for early visual areas. Rather than being the starting point as for PPCs, neural variability in sampling-based models emerges as a consequence of any uncertainty in the represented posterior. Importantly, samples have the same domain as the latents and do not normally relate to either log probability or probability directly.

This paper will proceed as illustrated in Figure 1: First, we will define a simple linear Gaussian image model as has been used in previous studies. Second, we will show that samples from this model approximate an exponential family with linear sufficient statistics. Third, we will relate the implied PPC, in particular the kernels, $\mathbf{h}(s)$, to the projective fields in our image model. Fourth, we will discuss the role of nuisance variables in our model. And finally, we will show that under assumption of binary latent in the image model, neural firing rates are both proportional to probability (of presence of a given image element) *and* log probability (of implicitly encoded variables like orientation).

## 2 A neural sampling-based model

We follow previous work in assuming that neurons in primary visual cortex (V1) implement probabilistic inference in a linear Gaussian model of the input image [14, 15, 12, 6, 10]:

$$P(\mathbf{I}|\mathbf{x}) = \mathcal{N}(\mathbf{I}; \mathbf{Ax}, \sigma_x^2 \mathbb{1}) \tag{2}$$

where $\mathcal{N}(y; \mu, \Sigma)$ denotes the probability distribution function of a normal random variable (mean $\mu$ and covariance $\Sigma$) evaluated at $y$, and $\mathbb{1}$ is the identity matrix. The observed image, $\mathbf{I}$, is

drawn from a Normal distribution around a linear combination of the projective fields ($\mathbf{PF}_n$), $\mathbf{A} = (\mathbf{PF}_1, \ldots, \mathbf{PF}_N)$ of all the neurons $(1, \ldots, N)$ weighted by their activation (responses), $\mathbf{x} = (x_1, \ldots, x_N)^\top$. The projective fields can be thought of as the brain's learned set of basis functions over images. The main empirical justification for this model consists in the fact that under the assumption of a sparse independent prior over $\mathbf{x}$, the model learns projective field parameters that strongly resemble the localized, oriented and bandpass features that characterize V1 neurons when trained on natural images [14, 6]. Hoyer & Hyvarinen (2003) proposed that during inference neural responses can be interpreted as samples in this model. Furthermore, Orban et al. (2016) showed that samples from a closely related generative model (Gaussian Scale Mixture Model, [24]) could explain many response properties of V1 neurons beyond receptive fields. Since our main points are conceptual in nature, we will develop them for the slightly simpler original model described above.

Given an image, $\mathbf{I}$, we assume that neural activities can be thought of as samples from the posterior distribution, $\mathbf{x}^{(i)} \sim p(\mathbf{x}|\mathbf{I}) \propto p(\mathbf{I}|\mathbf{x})p_{\text{brain}}(\mathbf{x})$ where $p_{\text{brain}}(\mathbf{x})$ is the brain's prior over $\mathbf{x}$. In this model each population response, $\mathbf{x} = (x_1, \ldots, x_N)^\top$, represents a sample from the brain's posterior belief about $\mathbf{x}|\mathbf{I}$. Each $x_n$, individually, then represents the brain marginal belief about the intensity of the feature $\mathbf{PF}_n$ in the image. This interpretation is independent of any task demands, or assumptions by the experimenter. It is up to the experimenter to infer the nature of the variables encoded by some population of neurons from their responses, e.g. by fitting this model to data. In the next section we will show how these samples can also be interpreted as a population code over some experimenter-defined quantity like orientation (Figure 1).

## 3 Neural samples form a Probabilistic Population Code (PPC)

In many classic neurophysiology experiments [17], the experimenter presents images that only vary along a single experimenter-defined dimension, e.g. orientation. We call this dimension the quantity of interest, or $s$. The question is then posed, what can be inferred about $s$ given the neural activity in response to a single image representing $s$, $\mathbf{x} \sim p(\mathbf{x}|s)$. An ideal observer would simply apply Bayes' rule to infer $p(s|\mathbf{x}) \propto p(\mathbf{x}|s)p(s)$ using its knowledge of the likelihood, $p(\mathbf{x}|s)$, and prior knowledge, $p(s)$. We will now derive this posterior over $s$ as implied by the samples drawn from our model in section (2).

We assume the image as represented by the brain's sensory periphery (retinal ganglion cells) can be written as

$$p(\mathbf{I}|s) = \mathcal{N}(\mathbf{I}; \mathbf{T}(s), \sigma^2_{\text{exp}\rightarrow\text{brain}}\mathbb{1}) \tag{3}$$

where $\mathbf{T}$ is the experimenter-defined function that translates the scalar quantity of interest, $s$, into an actual image, $\mathbf{I}$. $\mathbf{T}$ could represent a grating of a particular spatial frequency and contrast, or any other shape that is being varied along $s$ in the course of the experimenter. We further allow for Gaussian pixel noise with variance $\sigma^2_{\text{exp}\rightarrow\text{brain}}$ around the template $\mathbf{T}(s)$ in order to model both external noise (which is sometimes added by experimentalists to vary the informativeness of the image) and noise internal to the brain (e.g. sensor noise).

Let us now consider a single neural sample $\mathbf{x}^{(i)}$ drawn from the brain's posterior conditioned on an image $\mathbf{I}$. From the linear Gaussian generative model in equation (2), the likelihood of a single sample is

$$p(\mathbf{I}|\mathbf{x}^{(i)}) = \mathcal{N}(\mathbf{I}; \mathbf{A}\mathbf{x}^{(i)}, \sigma^2_x\mathbb{1}).$$

The probability of drawing $t$ independent samples[2] of $\mathbf{x}$ is,

$$
\begin{aligned}
p(\mathbf{x}^{(1,2,\ldots,t)}|\mathbf{I}) &= \prod_{i=1}^{t} p(\mathbf{x}^{(i)}|\mathbf{I}) \\
&= \prod_{i=1}^{t} \frac{p(\mathbf{I}|\mathbf{x}^{(i)})p_{\text{brain}}(\mathbf{x}^{(i)})}{p_{\text{brain}}(\mathbf{I})}
\end{aligned}
$$

$$= \frac{1}{p_{\text{brain}}(\mathbf{I})^t} \prod_{i=1}^{t} p(\mathbf{I}|\mathbf{x}^{(i)}) p_{\text{brain}}(\mathbf{x}^{(i)})$$

Since the experimenter and brain have different generative models, the prior over the variables are dependent on the generative model that they are a part of (specified by the subscript in their pdf). Substituting in the Gaussian densities and combining all terms that depend on $\mathbf{x}$ but not on $\mathbf{I}$ into $\kappa(\mathbf{x}^{(1,2,\ldots,t)})$, we get

$$p(\mathbf{x}^{(1,2,\ldots,t)}|\mathbf{I}) = \kappa\left(\mathbf{x}^{(1,2,\ldots,t)}\right) \frac{1}{p_{\text{brain}}(\mathbf{I})^t} \mathcal{N}\left(\mathbf{I}; \mathbf{A}\bar{\mathbf{x}}, \frac{\sigma_x^2}{t}\mathbb{K}\right). \tag{4}$$

where $\bar{x} = \frac{1}{t}\sum_{1}^{t} x^{(i)}$ is the mean activity of the units over time.
We next derive the posterior over samples given the experimenter-defined stimulus $s$:

$$p(\mathbf{x}^{(1,2,\ldots,t)}|s) = \int p(\mathbf{x}^{(1,2,\ldots,t)}|\mathbf{I}) p(\mathbf{I}|s) \mathrm{d}\mathbf{I}$$

Substituting in our result from equation (4), we obtain

$$p(\mathbf{x}^{(1,2,\ldots,t)}|s) = \kappa\left(\mathbf{x}^{(1,2,\ldots,t)}\right) \int \frac{1}{p_{\text{brain}}(\mathbf{I})^t} \mathcal{N}\left(\mathbf{I}; \mathbf{A}\bar{\mathbf{x}}, \frac{\sigma_x^2}{t}\mathbb{K}\right) p(\mathbf{I}|s) \mathrm{d}\mathbf{I}.$$

Making use of equation (3) we can write

$$p(\mathbf{x}^{(1,2,\ldots,t)}|s) = \kappa\left(\mathbf{x}^{(1,2,\ldots,t)}\right) \int \frac{1}{p_{\text{brain}}(\mathbf{I})^t} \mathcal{N}\left(\mathbf{I}; \mathbf{A}\bar{\mathbf{x}}, \frac{\sigma_x^2}{t}\mathbb{K}\right) \mathcal{N}(\mathbf{I}; \mathbf{T}(s), \sigma_{\text{exp}\to\text{brain}}^2\mathbb{K}) \mathrm{d}\mathbf{I}$$

$$= \kappa\left(\mathbf{x}^{(1,2,\ldots,t)}\right) \mathcal{N}\left[\mathbf{T}(s); \mathbf{A}\bar{\mathbf{x}}, \left(\sigma_{\text{exp}\to\text{brain}}^2 + \frac{\sigma_x^2}{t}\right)\mathbb{K}\right] \ldots$$

$$\int \frac{1}{p_{\text{brain}}(\mathbf{I})^t} \mathcal{N}\left[\mathbf{I}; \frac{\mathbf{T}(s)\sigma_x^2 + \mathbf{A}\bar{\mathbf{x}}t\sigma_{\text{exp}\to\text{brain}}^2}{t\sigma_{\text{exp}\to\text{brain}}^2 + \sigma_x^2}, \frac{\sigma_x^2 \sigma_{\text{exp}\to\text{brain}}^2}{t\sigma_{\text{exp}\to\text{brain}}^2 + \sigma_x^2}\mathbb{K}\right] \mathrm{d}\mathbf{I}$$

As the number of samples, $t$, increases, the variance of the Gaussian inside the integral converges to zero so that for large $t$ we can approximate the integral by the integrand's value at the mean of the Gaussian. The Gaussian's mean itself converges to $\mathbf{A}\bar{\mathbf{x}}$ so that we obtain:

$$p(\mathbf{x}^{(1,2,\ldots,t)}|s) \approx \kappa\left(\mathbf{x}^{(1,2,\ldots,t)}\right) \mathcal{N}\left[\mathbf{T}(s); \mathbf{A}\bar{\mathbf{x}}, \sigma_{\text{exp}\to\text{brain}}^2\mathbb{K}\right] \frac{1}{p_{\text{brain}}(\mathbf{A}\bar{\mathbf{x}})^t}.$$

Applying Bayes' rule and absorbing all terms that do not contain $s$ into the proportionality we find that in the limit of infinitely many samples

$$p(s|\mathbf{x}^{(1,2,\ldots,t)}) \propto \mathcal{N}(\mathbf{T}(s); \mathbf{A}\bar{\mathbf{x}}, \sigma_{\text{exp}\to\text{brain}}^2\mathbb{K}) p_{\text{exp}}(s). \tag{5}$$

We can now rewrite this expression in the canonical form for the exponential family

$$p(s|\mathbf{x}^{(1,2,\ldots,t)}) \propto g(s)\exp(\mathbf{h}(s)^\top \bar{\mathbf{x}}) \quad \text{where} \tag{6}$$

$$g(s) = \exp\left(-\frac{\mathbf{T}(s)^\top \mathbf{T}(s)}{2\sigma_{\text{exp}\to\text{brain}}^2}\right) p_{\text{exp}}(s) \quad \text{and} \tag{7}$$

$$\mathbf{h}(s) = \frac{\mathbf{T}(s)^\top \mathbf{A}}{\sigma_{\text{exp}\to\text{brain}}^2}. \tag{8}$$

If $\mathbf{x}^{(i)}$ is represented by neural responses (either spikes or instantaneous rates), $\bar{\mathbf{x}}$ becomes the vector of mean firing rates ($\mathbf{r}$) of the population up to time $t$. Hence, in the limit of many samples, the neural responses form a linear PPC (equation (1)).

**Finite number of samples**

The top row of Figure 2 shows a numerical approximation to the posterior over $s$ for the finite sample case and illustrates its convergence for $t \to \infty$ for the example model described in the previous section. As expected, posteriors for small numbers of samples are both wide and variable, and they get sharper and less variable as the number of samples increases (three runs are shown for each condition). Since the mean samples ($\bar{\mathbf{x}}$) only depends on the marginals over $\mathbf{x}$, we can approximate it using the mean field solution for our image model. The bottom row of Figure 2 shows the corresponding population responses: spike count for each neurons on the $y-$axis sorted by the preferred stimulus of each neuron on the $x-$axis.

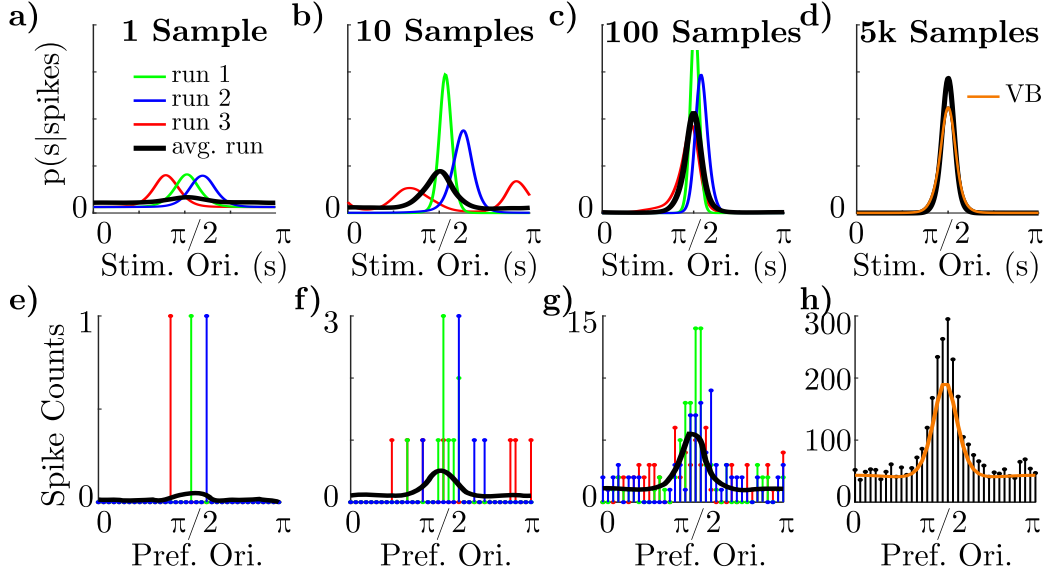

Figure 2: **a-c)** Posterior over $s$ for three runs (colored) and the expected posterior across many runs (black) for increasing number of samples. **d)** All runs converge to the same posterior (black). Posterior decoded from a mean-field Variational Bayes (VB) approximation to asymptotic firing rates in orange. **e-h)** Same simulations as in **a-d** but now plotting population spike counts sorted by each neuron's preferred orientation. Note that the counting window scales with the number of samples across panels. Panel **h** shows VB approximation to asymptotic firing rates in orange.

**Interpretation of the implied PPC**

The relationships that we have derived for $g(s)$ and $\mathbf{h}(s)$ (equations (7-8)) provide insights into the nature of the PPC that arises in a linear Gaussian model of the inputs. A classic stimulus to consider when probing and modeling neurons in area V1 is orientation. If the presented images are identical up to orientation, and if the prior distribution over presented orientations is flat, then $g(s)$ will be constant. Equation (7) shows how $g(s)$ changes as either of those conditions does not apply, for instance when considering stimuli like spatial frequency or binocular disparity for which the prior significantly deviates from constant. More interestingly, equation (8) tells us how the kernels that characterize how each neuron's response contribute to the population code over $s$ depends both on the used images, $\mathbf{T}(s)$, and the projective fields, $\mathbf{PF}_n$, contained in $\mathbf{A}$. Intuitively, the more $\mathbf{T}(s)^\top \mathbf{PF}_n$ depends on $s$, the more informative is that neuron's response for the posterior over $s$. Interestingly, equation (8) can be seen as a generalization from a classic feedforward model consisting of independent linear-nonlinear-Poisson (LNP) neurons in which the output nonlinearity is exponential, to a non-factorized model in which neural responses are generally correlated. In this case, $\mathbf{h}(s)$ is determined by the projective field, rather than the receptive field of a neuron (the receptive field, RF, being the linear image kernel in an LNP model of the neuron's response). It has been proposed that each latent's sample may be represented by a linear combination of neural responses [23], which can be incorporated into our model with $\mathbf{h}(s)$ absorbing the linear mapping.

Importantly, the kernels, $\mathbf{h}(s)$, and hence the nature of the PPC changes both with changes in the experimenter-defined variable, $s$ (e.g. whether it is orientation, spatial frequency, binocular disparity, etc.), and with the set of images, $\mathbf{T}(s)$. The $\mathbf{h}(s)$ will be different for gratings of different size and spatial frequency, for plaids, and for rotated images of houses, to name a few examples. This means that a downstream area trying to form a belief about $s$ (e.g. a best estimate), or an area that is combining the information contained in the neural responses $\mathbf{x}$ with that contained in another population (e.g. in the context of cue integration) will need to learn the $\mathbf{h}(s)$ separately for each task.

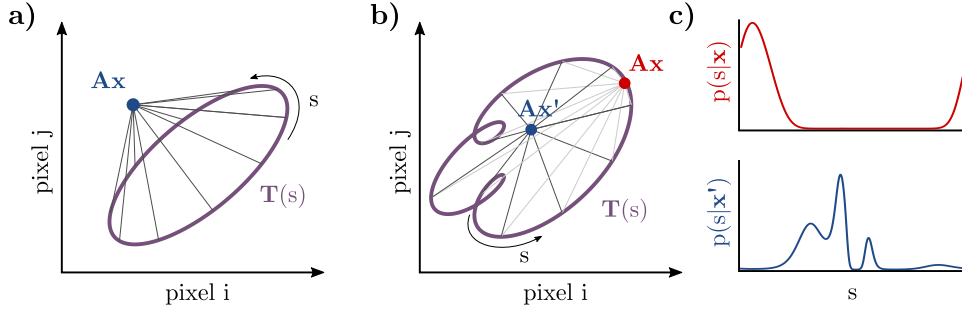

Figure 3: The link between $s$ and $\mathbf{x}$ is provided by the likelihood defined in image space. **a)** A manifold defined by $\mathbf{T}(s)$ is shown in the space of two example pixels. The likelihood of any stimulus $s$ for a particular sample $\mathbf{x}$ is related to the distance of that $\mathbf{x}$ projected into image space, $\mathbf{Ax}$, and $s$, projected into image space, $T(s)$ (up to $\sigma_{\text{exp}\rightarrow\text{brain}}$ noise). **b)** Same as **a**, but for a more complicated manifold. (Illustration, but note that rotating even a simple grating looks similar to the manifold shown here, not that in **a**.) The location of $\mathbf{Ax}$ in this space determines the relative heights of the multiple peaks of the implied posterior over $s$, shown in panel **c**.

**Multimodality of the PPC**

Useful insights can be gained from the fact that – at least in the case investigated here — the implied PPC is crucially shaped by the distance measure in the space of sensory inputs, $\mathbf{I}$, defined by our generative model (see equation 3). Figure 3 illustrates this dependence in pixel space: the posterior for a given value of $s$ is monotonically related to the distance between the image "reconstructed" by the mean sample, $\bar{\mathbf{x}}$, and the image corresponding to that value of $s$. If this reconstruction lies close enough to the image manifold defined by $\mathbf{T}(s)$, then the implied posterior will have a local maximum at the value for $s$ which corresponds to the $\mathbf{T}(s)$ closest to $\mathbf{A}\bar{\mathbf{x}}$. Whether $p(s|\mathbf{x}^{(1)}, \ldots, \mathbf{x}^{(t)})$ has other local extrema depends on the shape of the $\mathbf{T}(s)-$manifold (compare panels **a** and **b**). Importantly, the relative height of the global peak compared to other local maxima will depend on two other factors: (a) the amount of noise in the experimenter-brain channel, represented by $\sigma_{\text{exp}\rightarrow\text{brain}}$, and (b) how well the generative model learnt by the brain can reconstruct the $\mathbf{T}(s)$ in the first place. For a complete, or overcomplete model, for instance, $\mathbf{A}\bar{\mathbf{x}}$ will exactly reconstruct the input image in the limit of many samples. As a result, the brain's likelihood, and hence the implied posterior over $s$, will have a global maximum at the corresponding $s$ (blue in Figure 3B). However, if the generative model is undercomplete, then $\mathbf{A}\bar{\mathbf{x}}$ may lie far from the $\mathbf{T}(s)$ manifold and in fact be approximately equidistant to two or more points on $\mathbf{T}(s)$ with the result that the implied posterior over $s$ becomes multimodal with the possibility that multiple peaks have similar height. While V1's model for monocular images is commonly assumed to be complete or even overcomplete [25], it may be undercomplete for binocular images where large parts of the binocular image space do not contain any natural images. (Note that the multimodality in the posterior over $s$ discussed here is independent of any multimodality in the posterior over $\mathbf{x}$. In fact, it is easy to see that for an exponential prior and Gaussian likelihood, the posterior $p(\mathbf{x}|\mathbf{I})$ is always Gaussian and hence unimodal while the posterior over $s$ may still be multimodal.)

**Dissociation of neural variability and uncertainty**

It is important to appreciate the difference between the brain's posteriors over $\mathbf{x}$, and over $s$. The former represents a belief about the intensity or absence/presence of individual image elements in the input. The latter represents implicit knowledge about the stimulus that caused the input given the neural responses. Neural variability, as modeled here, corresponds to variability in the samples $\mathbf{x}^{(i)}$ and is directly related to the uncertainty in the posterior over $\mathbf{x}$. The uncertainty over $s$ encoded by the PPC, on the other hand, depends on the samples only through their *mean, not their variance*. Given sufficiently many samples, the uncertainty over $s$ is only determined by the noise in the channel between experimenter and brain (modeled as external pixel noise plus pixel-wise internal sensor noise added to the template, $\mathbf{T}(s)$). This means that an experimenter increasing uncertainty over $s$ by increasing external noise should *not* necessarily expect a corresponding increase in neural variability.

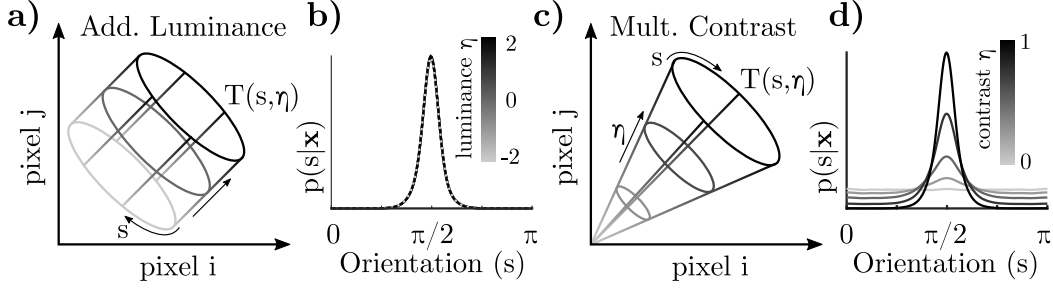

Figure 4: Illustration of the effect of two nuisance variables: luminance **(a-b)** and contrast **(c-d)** on image **(a,c)**, and on the corresponding posteriors over $s$ **(b,d)**. While the posterior is invariant to luminance, it depends contrast.

**Nuisance variables**

So far we have ignored the possible presence of nuisance variables beyond individual pixel noise. Such nuisance variables can be internal or external to the brain. Relevant nuisance variables when considering experiments done on V1 neurons include overall luminance, contrast, phases, spatial frequencies, etc (for an illustration of the effect of luminance and contrast see Figure 4). An important question from the perspective of a downstream area in the brain interpreting V1 responses is whether they need to be inferred separately and incorporated in any computations, or whether they leave the PPC *invariant* and can be ignored.

For any external nuisance variables, we can easily modify the experimenter's model in equation (3) to include a nuisance variable $\eta$ that modifies the template, $\mathbf{T}(s, \eta)$, and hence, the brain's observation, $\mathbf{I}$. This dependency carries through the derivation of the PPC to the end, such that

$$g(s, \eta) = \exp\left(-\frac{\mathbf{T}(s,\eta)^\top \mathbf{T}(s,\eta)}{2\sigma^2_{\mathrm{exp}\to\mathrm{brain}}}\right) p_{\mathrm{exp}}(s) \quad \text{and } \mathbf{h}(s, \eta) = \frac{\mathbf{T}(s,\eta)^\top \mathbf{A}}{\sigma^2_{\mathrm{exp}\to\mathrm{brain}}}. \tag{9}$$

As long as $\mathbf{T}(s, \eta)^\top \mathbf{T}(s, \eta)$ are separable in $s$ and $\eta$, the nuisance's parameter influence on $g$ can be absorbed into the proportionality constant. This is clearly the case for the contrast as nuisance variable as discussed in Ma et al. (2006), but in general it will be under the experimenter's control of $\mathbf{T}$ whether the separability condition is met. For the PPC over $s$ to be invariant over $\eta$, additionally, $\mathbf{h}(s)$ needs to be independent of $\eta$. For a linear Gaussian model, this is the case when the projective fields making up $\mathbf{A} = (\mathbf{PF}_1, \ldots, \mathbf{PF}_n)$ are either invariant to $s$ *or* to $\eta$. For instance, when $\mathbf{A}$ is learnt on natural images, this is usually the case for overall luminance (Figure 4a) since one projective field will represent the DC component of any input image, while the other projective fields average to zero. So while $\mathbf{T}(s, \eta)^\top \mathbf{PF}$ for the projective field representing the DC component will depend on the image's DC component (overall luminance), it does not depend on other aspects of the image (i.e. $s$). For projective fields that integrate to zero, however, $\mathbf{T}(s, \eta)^\top \mathbf{PF}$ is independent of $\eta$, but may be modulated by $s$ (e.g. orientation if the projective fields are orientation-selective).

The original PPC described by Ma et al. (2006) was shown to be contrast-invariant since both the "tuning curve" of each neuron, relating to $\mathbf{T}(s, \eta)^\top \mathbf{PF}$ in our case, and the response variance (taking the place of $\sigma^2_{\mathrm{exp}\to\mathrm{brain}}$) were assumed to scale linearly with contrast (in line with empirical measurements). For our model, we assumed that $\sigma_{\mathrm{exp}\to\mathrm{brain}}$ was independent of the input, and hence, the $\mathbf{T}$ are not invariant to contrast. However, since the noise characteristics of the brain's sensory periphery (included as sensor noise in our $\sigma_{\mathrm{exp}\to\mathrm{brain}}$ term) generally depend on the inputs, it remains a question for future research whether more realistic assumptions about the sensory noise imply an approximately invariant PPC over $s$. [3]

Generally speaking, the nature of the PPC will depend on the particular image model that the brain has learnt. For instance, numerical results by Orban et al. (2016) suggest that explicitly including

a contrast variable in the image model (Gaussian Scale Mixture, [24]) implies an approximately contrast-invariant PPC over orientation, but how precise and general that finding is, remains to be seen analytically.

# 4 Neurons simultaneously represent both probability & log probabilities

Taking the log of equation 6 makes it explicit that the neural responses, $\mathbf{x}$, are linearly related to the log posterior over $s$. This interpretation agrees with a long list of prior work suggesting that neural responses are linearly related to the logarithm of the probabilities that they represent. This contrasts with a number of proposals, starting with Barlow (1969) [1], in which neural responses are proportional to the probabilities themselves (both schemes are reviewed in [20]). Both schemes have different advantages and disadvantages in terms of computation (e.g. making multiplication and addition particularly convenient, respectively) and are commonly discussed as mutually exclusive.

While in our model, with respect to the posterior over $\mathbf{x}$, neural responses generally correspond to samples, i.e. neither probabilities nor log probabilities, they do become proportional to probabilities for the special case of binary latents. In that case, on the time scale of a single sample, the response is either 0 or 1, making the firing rate of neuron $i$ proportional to its marginal probability, $p(\mathbf{x}_n|\mathbf{I})$. Such a binary image model has been shown to be as successful as the original continuous model of Olshausen & Field (1996) in explaining the properties of V1 receptive fields [11, 6], and is supported by studies on the biological implementability of binary sampling [7, 18].

In sum, for the special case of binary latents, responses implied by our neural sampling model are at once proportional to probabilities (over $\mathbf{x}_n$), and to log probabilities (over $s$).

# 5 Discussion

We have shown that sampling-based inference in a simple generative model of V1 can be interpreted in multiple ways, some previously discussed as mutually exclusive. In particular, the neural responses can be interpreted both as samples from the probabilistic model that the brain has learnt for its inputs *and* as parameters of the posterior distribution over any experimenter-defined variables that are only implicitly encoded, like orientation. Furthermore, we describe how both a log probability code as well as a direct probability code can be used to describe the very same system.

The idea of multiple codes present in a single system has been mentioned in earlier work [23, 5] but we make this link explicit by starting with one type of code (sampling) and showing how it can be interpreted as a different type of code (parametric) depending on the variable assumed to be represented by the neurons. Our findings indicate the importance of committing to a model and set of variables for which the probabilities are computed when comparing alternate coding schemes (e.g. as done in [9]).

Our work connects to machine learning in several ways: (1) our distinction between explicit variables (which are sampled) and implicit variables (which can be decoded parametrically) is analogous to the practice of re-using pre-trained models in new tasks, where the "encoding" is given but the "decoding" is re-learned per task. Furthermore, (2) the nature of approximate inference might be different for encoded latents and for other task-relevant decoded variables, given that our model can be interpreted either as performing parametric or sampling-based inference. Finally, (3) this suggests a relaxation of the commonplace distinction between Monte-Carlo and Variational methods for approximate inference [22]. For instance, our model could potentially be interpreted as a mixture of parametric distributions, where the parameters themselves are sampled.

We emphasize that we are not proposing that the model analyzed here is the best, or even a particular good model for neural responses in area V1. Our primary goal was to show that the same model can support multiple interpretations that had previously been thought to be mutually exclusive, and to derive analytical relationships between those interpretations.

The connection between the two codes specifies the dependence of the PPC kernels on how the image manifold defined by the implicit variable interacts with the properties of the explicitly represented variables. It makes explicit how infinitely many posteriors over implicit variables can be "decoded" by taking linear projections of the neural responses, raising questions about the parsimony of a description of the neural code based on implicitly represented variables like orientation.

We also note that the PPC that arises from the image model analyzed here is not contrast invariant like the one proposed by Ma et al. (2006), which was based on the empirically observed response variability of V1 neurons, and the linear contrast scaling of their tuning with respect to orientation. Of course, a linear Gaussian model is insufficient to explain V1 responses, and it would be interesting to derive the PPC implied by more sophisticated models like a Gaussian Scale Mixture Model [24] that is both a better model for natural images, enjoys more empirical support and, based on numerical simulations, may approximate a contrast-invariant linear PPC over orientation [16].

Finally, a more general relationship between the structure of the generative model for the inputs, and the invariance properties of PPCs empirically observed for different cortical areas, may help extend probabilistic generative models to higher cortical areas beyond V1.

**Acknowledgments**

This work was supported by NEI/NIH awards R01 EY028811 (RMH) and T32 EY007125 (RDL), as well as an NSF/NRT graduate training grant NSF-1449828 (RDL, SS).

## Footnotes

[2]Depending on how the samples are being generated, consecutive samples are likely to be correlated to some degree. However, the central result derived in this section which is valid for infinitely many samples still holds due to the possibility of thinning in this case. Only for the finite sample case will autocorrelations lead to deviations from the solutions here

[3] In contrast to the interpretation of Ma et al. (2006), where contrast invariance is the result of a combination of mean response scaling and response variance scaling, in our case it would be a combination of the "feedforward" part of the mean response scaling and the scaling of the variability of the *inputs*.

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
