[Reviews · NeurIPS 2018]

Reviewer 1



The authors start by assuming that: (1) neural responses x are samples from p(x|Image), and (2) the brain already has (or has pre-learned) a linear Gaussian generative model of images given responses i.e. p(Image|x) is N(Ax,noiseSD). Using these, they derive that the log posterior over some experimenter defined variable s (that generated the image using an arbitrary function) is a linearly weighted sum of neural responses x; i.e. the neural responses form a probabilisitic population code (PPC) which can be linearly decoded to give the posterior over any experimenter defined variable that generated the images. The authors thus show that the sampling vs PPC hypotheses are not disjoint, but can actually co-exist by properly defining what is being sampled and what is coded in the PPC. This is a very significant result and definitely deserves to be widely disseminated in the community. Thus I recommend this work to be accepted at NIPS, after these corrections. major: l 111: If I follow the math correctly, then in the second equality after line 111, there seems to be an extra factor of a normal distribution function with three dots trailing. While the paper is clear, the authors must put in more effort in proof-reading their texts, so as to not burden reviewers with a huge number of trivial corrections as below! Laxity here also raises doubts on the rigour in the main results ... minor: l 54: "and do not normally related to either log probability or probability directly." l 86: "presents a images" l 71-72: "Furthermore, and Orban et al. (2016)" l 90: p(s|x) not p(s|r) l 98: Equation 3 and the one below. How does the Normal distribution function have 3 arguments here compared to two arguments earlier? What is the symbol at the end that looks like an Identity symbol? pg 3 footnote: "infinitely samples" l 106: "we get Dr" l 106: equation 4: \bar{x} should be defined. l 165: "will have a global maximum for at the corresponding" l 199: "it will be under the experimenter’s control of T whether the separability condition is met" l 200: " to be invariance over " l 230-231: "generally correspond samples" l 232: "a binary latents" l 228-229: "e.g. making addition and multiplication particularly, respectively"

Reviewer 2



REPLY TO THE AUTHORS' FEEDBACK ============================== Thank for very much for the response and the very concrete description of changes. I can picture how the updated manuscript looks. With the presentation issues out of the way, I raised my score to 9 points, i.e., a strong vote for acceptance. Just for completeness, allow me to clarify the "Strictly speaking the statement is correct” (L163ff). In the idealized experiment considered in Fig 3, there are two sources of stochasticity: The neural samples x|I and the images I|T(s). For the main result ( eqs (6)-(8) ), the images I have been marginalized out. Thus, I understand the discussion around L163ff such that the network can describe the _average_image_T(s)_ well in case of a complete model A. The term "image" alone (without an attribute such as "average") is defined as the noisy realization I, according to line 76. Therefore, the sentence "Ax̄ will exactly reconstruct the input image" [...the limit can only refer to the x-samples.] describes the network's ability to capture also the noise-realization in I. For a complete model A, this is indeed the case. So, strictly speaking the statement is correct. But the discussion, as I understand it, refers to the average image, or prototype, T(s). Anyhow, "Congratulations!" to a great piece of science. In light of the topic's relevance, I recommend this paper for a talk. SUMMARY ======= The authors investigate the relation between sample-based codes and probabilistic populations codes (PPCs), two prevalent -- and often competing -- theoretical interpretations of neural activity for representing Bayesian computations in brain circuits. As the main result, the authors reconcile the two interpretations by showing analytically and in computer simulations how a sample-based code can yield a PPC code in the limit of many samples. For this, the authors set out with two independent, widely accepted modeling assumptions: (1) A linear Gaussian generative model of the input with neural sampling activations as latent variables. (2) An almost unrestricted Gaussian generative model of the input conditioned on a relevant stimulus parameter. Marginalizing over inputs followed by some mathematical massaging leads to the main result: a PPC form for p( stimulus parameter | neural activations ). The paper then discusses some often debated topics (multimodality, nuisance variables, linear vs log-probabilities) from the unified standpoint of their model. QUALITY: 6 ========== + The introduction is very thoughtfully written: precisely carving out the assumptions and differences of the two compared interpretations. + This is one of the few papers on this topic that do not intend to bolster up one interpretation over the other. - It would be interesting to see if the derivation holds for more general (exponential family) generative models beyond linear Gaussian. + I think that the derivation is correct, once the reader... - corrected the many small mistakes in the equations, - skipped over copy&paste ruins (entire lines could be removed), - ignored unused and false assumptions, - guessed the definition of multiple undefined important quantities. These issues impact both quality and clarity of the manuscript. Please see below for details. CLARITY: 3 ========== The presentation of the central derivation in sections 2 and 3 leaves (politely speaking) the impression of a very hasty writing process. Besides the above-mentioned errors, there is no high-level outline of the argumentation. Figure 1 could help explaining this, but the figure is never referred to in the main text. And the argumentation would be so simple: [A] From eq. (2) and assuming conditionally independent samples (x^i ⊥ x^j | I) follows eq (4). [B] From eq. (4) and the stimulus model ( what should be eq. (3); currently unnumbered ) follows eq. (5) in the limit of many samples. [C] Eq (5) can be cast into the canonical PPC form eq (6) - (8) by identifying r with x^[1:t]. The last step, identifying r with x^[1:t], is completely omitted in the manuscript, but crucial since this connects the quantities of the two interpretations. Overall, the clarity of presentation is the major weakness of the manuscript. Should the manuscript be rejected for this reason (which is not unlikely), I would like to encourage the authors to get the maths correct and resubmit to a leading computational neuroscience journal (e.g., PLoS CompBiol or PNAS). ORIGINALITY: 8 ============== The investigated question is around already for several years. The paper is original in that someone finally worked out the details of the maths of both representations. This is a very important contribution which requires a fine understanding of both theories. A big "Thanks" to the authors for this! Yet, I think that two relevant papers are missing: - Savin, C., Deneve, S. Spatio-temporal representations of uncertainty in spiking neural networks, Advances in Neural Information Processing Systems 27, 2014. This paper relaxes the assumption of a one-to-one correspondence between neurons and latent variables for sampling. - Bill J, Buesing L, Habenschuss S, Nessler B, Maass W, et al. (2015) Distributed Bayesian Computation and Self-Organized Learning in Sheets of Spiking Neurons with Local Lateral Inhibition. PLOS ONE In the discussion section, a similar relation between sampling and PPCs is outlined, for the case of a generative mixture model instead of the linear Gaussian model considered here. SIGNIFICANCE: 10 ================ Sampling vs PPC representations is a hotly debated topic for years, staging the main track talks of the most important computational neuroscience conferences. Reconciling the two views is highly relevant for anyone who studies neural representations of uncertainty. CONCLUSION ========== This is one of the papers, I feel thankful that it has finally been written (and researched, of course). Unfortunately, the paper suffers from an unjustifiable amount of typos and errors in the maths. If accepted to NIPS, I would suggest a talk rather than a poster. Overall, I tend to recommend acceptance, but I could perfectly understand if the quality of presentation does not merit acceptance at NIPS. Due to the very mixed impression, I assigned a separate score to each of the four reviewing criteria. The geometric mean leads to the suggested overall score 6/10. SPECIFIC COMMENTS ================= - L163ff: Strictly speaking the statement is correct: A complete basis A permits to perfectly model the noise in I \sim N(T(s), noise). But I think this is not what the authors mean in the following: The example addresses how well Ax approximates the template T(s) without a noise sample. If so, the discussion should be corrected. Errors in the derivation / equations ------------------------------------ Since I walked through the derivation anyway, I also include some suggestions. - Eq (1): The base measure is missing: ... + f(s); alternatively write p(s|r) \propto exp(...) to match eq (6) of the main result. - L66: Missing def. of PF_i; required e.g. in line 137 - L90: p(s|x) not r - L90 and general: The entire identification of r <--> multiple samples x^[1:t] is missing, but crucial for the paper. - Eq after L92: This should be eq (3). These are the "forward physics" from stimulus generation to spikes in the retina. The name p_exp(I|s) is used later in L109ff, so it should be assigned here. - Eq (3): Copy of the previous eq with wrong subscript. Could be deleted, right? - Eq after L106: Missing def of \bar x = 1/t \sum x^i (important: the mean is not weighted by p(x^i) ) - Eq after L107: p_exp(x|s) is not defined and also not needed: delete equation w/o replacement - L108 and L109: The equality assumption is undefined and not needed. Both lines can be deleted. The next eq is valid immediately (and would deserve a number). - L117 and eq 6: At some point r (from eq 1) and x^[1:t] must be identified with another. Here could be a good point. Section 4 could then refer to this in the discussion of binary x^i and longer spike counts r. Minor Points ------------ - Figure 1 is not referred to in the text. - L92: clarify "periphery" (e.g. retinal ganglion cells). This is an essential step, bringing the "pixel image" into the brain, and would deserve an extra sentence. - L113: The Gaussian converges to a dirac-delta. Then the integration leads to the desired statement. Typos ----- - Figure 1, binary samples, y_ticks: x_n (not N) - Caption Fig 1: e.g.,~ - L48: in _a_ pro... - L66: PF_n (not 2) - L71: ~and~ - L77: x^(i) not t [general suggestion: use different running indices for time and neurons; currently both are i] - Almost all the identity matrices render as "[]" (at least in my pdf viewer). Only in the eq after L92, it is displayed correctly. - Footnote 1: section_,_ valid for infinitely _many_ samples_,_ ... - L106: Dr - Caption Fig 2: asymptotic - L121: section 3 (?, not 4) - L126: neuron~s~ - Figure 3C: ylabels don't match notation in 3B (x and x'), but are illustrations of these points - L165: ~for~ - L200: invariant - L220: _is_ - L229: _easy_ or _simple_ - L230: correspond _to_ - L232: ~a~ - L242: ~the~

Reviewer 3



The authors tackle an important problem in systems neuroscience, namely whether and how neural activity can be interpreted as implementing probabilistic inference. In particular, two theories have been put forward in the past, probabilistic inference and sampling-based approaches. The authors claim to be able to unify the two approaches, thus providing an integrated account of probabilistic inference with neural populations. To this end, the authors study a relatively simple Gaussian image model, where an image is represented in a neural population with independent Gaussian noise. From this, they compute the posterior and show that the mean of samples from the posterior corresponds to a linear PPC. The interpretation of this PPC makes several interesting prediction, e.g. that adding noise to an image does not necessarily imply more variance in the spike counts, as the uncertainty of the representation is related to the mean, not the variance of the samples. While the paper touches on an important topic, and the calculations seem largely correct, the relationship of the model to neural data could have been worked out more, e.g. with more concrete examples. As it is, the model does not seem particularly biologically plausible, as common ingredients such as Poisson noise are missing. Even illustrating a few of the main ingredients and how they work together would have been helpful. The analysis of the model presented in Fig. 3/4 seems to be more illustrative than actual results, here a more in depth interpretation would be helpful. 106: extra Dr Fig 2: asymptotive -> asymptotic 125: left column -> bottom row?